# UNVEILING THE DYNAMICS OF TRANSFER LEARNING REPRESENTATIONS

**Thomas Goerttler & Klaus Obermayer**
Neural Information Processing
Technische Universität Berlin
Berlin, Germany
{thomas.goerttler,klaus.obermayer}@tu-berlin.de

## ABSTRACT

Representation similarity analysis is used to analyze the dynamics of neural networks. When used to measure the importance of layers in fine-tuning, it is revealed that there is less representation change in early layers than in later layers, which supports freezing early layers during fine-tuning. In this paper, we want to discuss how we can interpret these similarity scores of the representations. We argue that the scalar value of similarity scores between representations of trained and untrained networks should not be interpreted directly. In addition, similarity values obtained by comparing learned representations to their initialized representation should not be compared across layers to judge their importance. Instead, the similarity scores should be proportioned to similar problems to be assessed appropriately. This can be done by a controlled randomization of the dataset, which covers the spectrum from original to random. We find out that the representation change depends on the size of the training data, its structure, and - if pre-trained - how close it is to the pre-trained task. If a dataset does not have a meaningful hierarchical structure, smaller networks tend to *unlearn* the knowledge of the pre-trained network. In contrast, larger networks still use their learned capabilities.

## 1 INTRODUCTION

Despite the success of transfer learning, the underlying reasons for the effectiveness of transfer learning remain partially undisclosed. It is widely believed that the initial layers of the network focus on low-level features (e.g., edges and blobs), gradually progressing to high-level features (e.g., objects) (Zhuang et al., 2021). This intuition has led to the common practice of fine-tuning only the final layers, specifically the fully connected ones, for adaptation to new yet analogous tasks, also called linear probing (Kumar et al., 2022). This process also serves as implicit regularization and prevents overfitting by reducing parameters.

Existing research, exemplified by studies like Raghu et al. (2020) and Oh et al. (2021), uses representation similarity analysis, a popular technique in deep neural network analysis, to shed light on the behavior of these early layers during fine-tuning. A smaller representation change in early layers supports freezing this part. However, we believe that the interpretation of representation similarity analyses requires careful consideration, particularly when analyzing the similarity of representations to their initialization and comparing it to other layers.

Representation similarity analysis, as proposed by Kriegeskorte et al. (2008); Kornblith et al. (2019), is an attractive technique for deep neural network analysis (Raghu et al., 2021; Beyer et al., 2023; Sucholutsky et al., 2023) and can be used to compare representations of layers from the same networks, from the same networks after different training steps but also from different networks as long as the representations are obtained from the same input data (Kornblith et al., 2019; Raghu et al., 2021; Kornblith et al., 2021; Beyer et al., 2023).

Representation analysis can also be applied to measure the *representation change* during fine-tuning. For example, Raghu et al. (2020); Oh et al. (2021) employ CKA scores to emphasize layer importance, inferring that a minimal representation change indicates less crucial layers. But, Goerttler & Obermayer (2022) reveal that this behavior is pervasive in deep neural networks, occurring during

pre-training and especially in scenarios with limited data. We confirm these observations with experiments on larger networks and think that comparing representation change across different layers is intricate, given the direct dependency of late layers on early layer activations. This effect is even more significant when we only fine-tune on a very small dataset, e.g., in episodic few-shot learning (Goerttler & Obermayer, 2021).

Therefore, it is necessary to take care when interpreting similarity scores, particularly concerning their representation similarity to their initial representation. The goal of the paper is to address the question of whether representation similarity measures can be used to quantify the importance of a layer in fine-tuning, and we want to assess if early layers are beneficial for fine-tuning. We use a gradual way of randomizing the labels of the dataset so that we can compare learning strategies and memorization (Goerttler & Obermayer, 2022). We analyze representation obtained by models having initialized, pre-trained, and fine-tuned weights for smaller networks (4conv) and larger networks (VGG, ResNet).

By analyzing the training pre-training deep network model, we confirm that early layers structurally change less than later ones, and this effect is even stronger if the fine-tuned dataset is smaller (Section 4). To investigate the representation change during fine-tuning, we apply pre-trained models to structured, unstructured, domain, and cross-domain tasks. We confirm that the data structure is learned in the early convolutional layers, whereas memorization causes less representation change. In addition, we reveal that if the transfer task is unstructured (nosier labels), the pre-trained knowledge is *unlearned* in smaller networks but interestingly used in larger networks. However, in a domain and cross-domain adaption, it exploits the knowledge also for smaller networks (Section 5).

Overall, we realize that similarity scores must be carefully interpreted because a seemingly small representation change of a layer might be small only due to its position in the network. Instead, it is better to compare it to similar problems with the same architecture to assess its significance.

## 2 RELATED WORK

Yosinski et al. (2014) reveal that first-layer features in these networks resemble Gabor filters or color blobs. They shed light on the generality and specificity of features at different layers, emphasizing a diminishing trend in transferability as layers progress, with lower layers demonstrating superior transferability (Yosinski et al., 2014). Providing a comprehensive overview of transfer learning, Kumar et al. (2022) find out that fine-tuning can distort pre-trained features which harms out-of-distribution performance. Hoffer et al. (2018) show that fixing the classifier is superior to training it. Episodic approaches to address few-shot learning, as outlined by Vinyals et al. (2016); Finn et al. (2017), aim to optimize for quick fine-tuning on very small datasets. Building upon these approaches, Raghu et al. (2020); Oh et al. (2021) extend the analysis by investigating representation similarity.

Representation similarity techniques are widely used in computational neuroscience and machine learning to analyze how similar two representations are. In neuroscience, RSA (representation similarity analysis) is used, e.g., to compare a computational or behavioral model with the brain response (Kriegeskorte et al., 2008). In deep learning, most often, centered kernel alignment (CKA) (Kornblith et al., 2019) is used to compare representations (Raghu et al., 2020; Oh et al., 2021; Kornblith et al., 2021; Neyshabur et al., 2020). Recently, Sucholutsky et al. (2023) provided a detailed analysis of representation alignment, aiming to reinforce the connection of tools in neuroscience, cognitive science, and machine learning. They distinguish between three types of representation similarity: measuring the degree of alignment between two systems as a dependent measure in an experiment (Kriegeskorte et al., 2008; Kornblith et al., 2019); bridging representational spaces to bring representations into a shared space to facilitate downstream applications (Gupta et al., 2017); increasing representational alignment to update the internal representation or measurement of one system to increase its alignment with another system (Tian et al., 2020).

## 3 REPRESENTATION SIMILARITY ANALYSIS OF DEEP NEURAL NETWORKS

Measuring the similarity between representations of deep neural networks serves as a crucial tool for addressing various inquiries within the field. Centered Kernel Alignment (CKA) has emerged as

the state-of-the-art method in deep learning for comparing these representations (Raghu et al., 2020; Oh et al., 2021; Kornblith et al., 2021; Neyshabur et al., 2020; Dwivedi et al., 2020; Sucholutsky et al., 2023). The methodology of CKA is defined as follows: Given matrices $X \in \mathbb{R}^{n \times k_1}$ and $Y \in \mathbb{R}^{n \times k_2}$ containing centered activation values of $k_1$ and $k_2$ neurons respectively, for $n$ instances, the objective is to derive a scalar $s(X, Y)$ enabling comparison of representations both within and across neural networks. This involves computing representation similarity matrices $XX^T$ and $YY^T$, which present the similarity of each pair of examples. Subsequently, the Hilbert-Schmidt Independence Criterion (HSIC) is employed, and the index is normalized to obtain a scalar value that describes the similarity of two representations, $X$ and $Y$. A similarity score approximating 1 indicates minimal representation change, while a score nearing 0 suggests significant representation change.

CKA can be used to compare representations, e.g., to measure the

- representation similarity of layers of the same network with each other,
- representation similarity of layers of the same network trained on different seeds, hyperparameters, or datasets[1],
- representation similarity of layers of two different networks,
- representation similarity of a layer across time,
- representation similarity of multiple layers w.r.t. its initialization.

Depending on the goal of the analysis, it makes sense to analyze the matrices of pair-wise scalars and compare multiple of them in relation to or only several pairs of similarity.

We compare the representation similarity of different layers compared to its initial representation and call it analogously to Oh et al. (2021) representation change. In our experiments, we focus on comparing the representation before training with the representation during and after training of the same layer as done by Raghu et al. (2020) and Oh et al. (2021). For every layer, there are only two representation similarity matrices to compare: the pre-initialed representation, which is the one obtained before training, and the resulting representation. These are aggregated layer-wise.

### 3.1 MODELS

In our experiments, we use three different architectures. First of all, the 4-conv architecture, which has been proposed by Vinyals et al. (2016) and also used by Raghu et al. (2020); Oh et al. (2021). The architecture consists of 4 modules with 3x3 convolutions and 64 filters. These are followed by batch normalization, a Rectified Linear Unit activation function, and a pooling layer (2x2). The output of the fourth block is flattened and fully connected with the output layer.

Secondly, we use VGG16, a convolutional neural network architecture proposed by the Visual Geometry Group (VGG) and detailed by Simonyan & Zisserman (2015). VGG16 stands out for its simplicity, comprising 16 weight layers, including 13 convolutional layers and 3 fully connected layers. Notably, each convolutional layer employs small 3x3 filters, and max-pooling is applied after the second, fourth, seventh, tenth, and thirteenth convolutional blocks. We use the VGG16 version, which includes batch normalization after the convolutional operation. This architecture has found widespread use in image classification tasks and remains a benchmark in the field.

The last network employed in our experiments is residual neural network (ResNet) (He et al., 2016). They consist of residual blocks, which contain a certain number (mostly 2 or 3) of convolutional layers and a skip connection, which forwards the identity, enabling the learning of residual functions. This architecture is particularly known for its effectiveness in training very deep networks. Our experiments use the Resnet18 architecture, which initiates with a convolutional layer featuring 64 filters, accompanied by batch normalization and ReLU activation. ResNet18 incorporates a series of residual blocks, introducing shortcut connections. Each residual block consists of two convolutional layers with batch normalization and ReLU activation and a skip connection that adds the output to the block's input. The architecture comprises four sets of residual blocks with varying filter numbers

---

[1]To meaningfully compare two representations, we have to compare the activation when the network is applied on the same dataset. While it technically works also if the size of the dataset is the same, its interpretation is not possible

(64, 128, 256, 512) and periodic spatial dimension reductions through strided convolutions or max-pooling. A global average pooling layer follows, averaging spatial dimensions and producing a single value for each feature map. The network concludes with a fully connected layer employing softmax activation for final classification probabilities.

## 3.2 DATASETS

We apply the network to several standard datasets in machine learning (ImageNet, CIFAR-10, and SVHN) and randomized versions for our analysis. We structurally add noise to the labels such that we cover the spectrum from original labels to fully randomized labels, not only in extreme situations but also in the partially randomized version (Section 3.3). We also shift the dataset such that, in theory, only the label assignment changes from training and testing (Section 3.4).

CIFAR-10, SVHN (Street View House Numbers), and ImageNet are well-known datasets widely utilized in the field of machine learning for image classification tasks. CIFAR-10 consists of 60,000 32x32 color images across ten classes, each containing 6,000 images, making it a benchmark for evaluating algorithms on relatively small and low-resolution images. SVHN, on the other hand, focuses on recognizing digits in natural scene images and consists of over 600,000 32x32 color images of house numbers extracted from Google Street View. ImageNet is a substantially larger dataset, containing millions of high-resolution images across 1,000 classes. It has been a cornerstone in the development and evaluation of deep neural networks, particularly convolutional neural networks (CNNs), due to its diverse and extensive image collection spanning a wide range of categories.

## 3.3 RANDOMIZING DATASETS

Having a labeled dataset with $D$ classes which are named with numbers from $0$ to $D-1$, Goerttler & Obermayer (2022) define a partially random label $y_d$ with a degree of randomness $d \in \{0, 1, ..., D-1\}$ as:

$$y_d = (y + Y) \mod D \tag{1}$$

where $Y$ is a random variable with a probability function of

$$Pr(y = Y) = \begin{cases} \dfrac{1}{d+1} & \text{if } y \in \mathbb{N}_0 \text{ and } y \leq d \\ 0 & \text{otherwise} \end{cases} \tag{2}$$

Each original label is uniformly distributed over $d+1$ labels, such that the closer $d$ is to $D$, the more random the labels are, and less structure remains in the problem. A choice of $d = 0$ is equivalent to the original dataset, whereas $d = D-1$ means that the dataset is completely random. In Figure 1, we see an example of CIFAR-10 with a random degree $d = 1$, where every class is assigned to two labels. Inversely, every label covers two classes. In Appendix A, we show more examples of random datasets.

## 3.4 SHIFTED DATASETS

In addition to randomizing the labels, we also perform experiments on shifted versions of the dataset. The idea is that we change the number every class is assigned to between training and testing so that only the label assignment has to change in the fine-tuning. The new labels $Y_s$ are defined as

$$y_S = (y + s) \mod D, \tag{3}$$

where $s \in \{1, ..., D-1\}$. This represents a similar dataset that lies in the same domain, but the novel labels have to be relearned. In Appendix A, we show an example of a shifted dataset. Since we depend on the shifting of the seed value, the shift is different from every run.

## 3.5 TRAINING SETUP

Every run, regardless of whether pre-training or fine-tuning, is repeated for five different seeds. In addition, we indicate the standard error of the results. We use the stochastic gradient descent with a learning rate of 0.001 and a momentum of 0.9. For all fine-tuned tasks, we used 5000 samples for

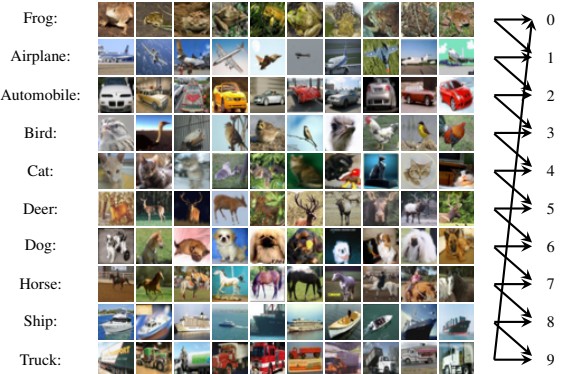

Figure 1: Example of the CIFAR-10 dataset with a random degree of $d = 1$, meaning that $50\%$ of a class is assigned to one number and the other $50\%$ to another class. Every label covers two different classes.

training and 5000 for testing. [2]. We pre-trained VGG16 and ResNet18 for 50 epochs each and the 4-conv architecture for 150 epochs. We then fine-tuned each model for an additional 25 epochs.

## 4 DEEP NEURAL NETWORKS DURING PRE-TRAINING

We first want to see how the representation change is during learning standard deep networks, which serve as a pre-trained model in transfer learning. This explains how learning random initialized networks affects the representation change and is a good reference for analyzing fine-tuned networks later.

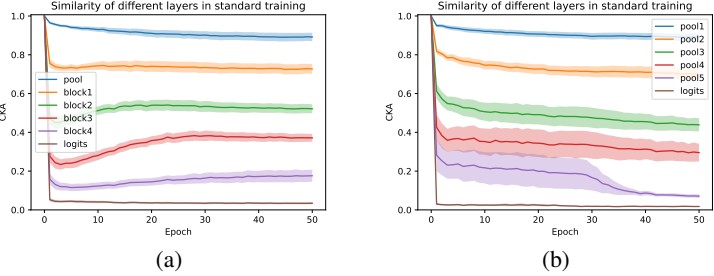

Figure 2: These figures show the similarity of the representations of the layers to its initial representation of ResNet18 applied on CIFAR-10 during training.

In our first experiment, we look at the representation change of the layers to the representation obtained after random initialization and control how it evolves during training (see Figure 2). We observe that the representations of early layers change less than the later ones during standard training. This emphasizes our position that less representation change in the early layer is common for neural networks.

Therefore, we have to be careful to conclude that smaller adaptations in early layers imply that they are not as beneficial or can be discarded during fine-tuning, as it has been done by Raghu et al. (2020)[3].

We think this can have several reasons:

---

[2]also for SVHN and ImageNet, although there is, in theory, more data available, we want to be consistent
[3]Raghu et al. (2020) also had other empirical reasons to suggest their approach, but partially justified it with the small representation change.

- The hierarchical structure of the input data, early layers tend to learn more basic and general features (fundamental aspects).
- The stacked architecture leads to a dependency between a succeeding layer and its preceding layer, affecting the evolution of representations throughout the network.
- Vanishing gradient problem.

We think that it is neither fair nor meaningful to compare the representation change (towards its initial representation) of different layers to each other, but it makes more sense to compare a layer to the same layer of similar problems. We therefore compare it to experiments where the dataset is similar but slightly more random. In this setup, we can observe how the representation changes with a decrease in randomness in the dataset. We apply our models to more randomized versions of our dataset, whose generation is described in Section 3.3.

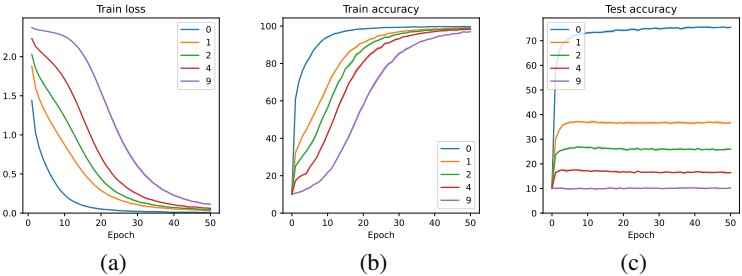

(a)      (b)      (c)

Figure 3: This figure depicts the training loss, training accuracy and the accuracy during testing of the application of ResNet18 on for degrees of randomness $d \in \{0, 1, 2, 4, 9\}$ of CIFAR-10.

### 4.1 DECREASE OF STRUCTURE IN THE TRAINING DATA

We want to observe the difference of the representation change while controlling the structure of the dataset. The more randomness to the labels is applied, the neural network can learn less knowledge, and it can only memorize the training data. Before we analyze the resulting similarities, we want to have a look at training and testing statistics. In Figure 3, we depict the statistics of ResNet18 when trained on CIFAR-10. The results of VGG16on CIFAR-10 can be seen in Appendix B. In Figure 3(a), we observe that if the data is more random and, therefore, has less structure, the loss decreases slowly. The overall problem is more difficult, and it requires memorizing the labels and not taking advantage of the shared structure in the data. Nevertheless, suppose the models are trained long enough. In that case, the training accuracy in the experiments on the randomized data is almost as high as when trained on the randomized data (see Figure 3(b)). The test accuracy in Figure 3(c) shows that the networks only learn meaningful knowledge if there is structure in the data.

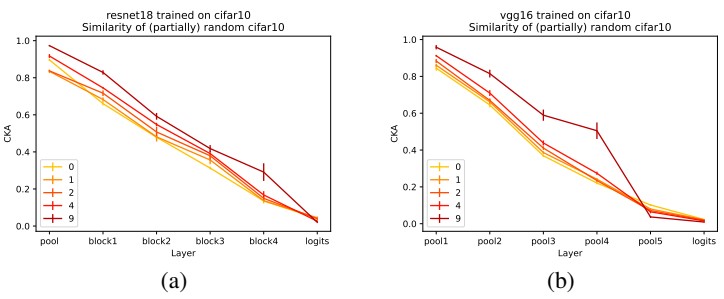

(a)        (b)

Figure 4: Representation change compared to the initial state of several layers of ResNet18 and VGG16 after training on CIFAR-10 with (partially) random labels ($d \in \{0, 1, 2, 4, 9\}$).

Looking at the representation change during the training, results of Figure 4, we observe that if the structure in the data decreases (higher degree of randomness), there is less similarity change overall.

This holds true, especially in the early layers. Only the last pooling layer of VGG16 (Figure 4(b)) it is the opposite, but the similarity scores are all below 0.1, which is close to not similar at all. Since the training accuracy is very good for all versions of datasets (see Figure 3(b)), we can conclude that memorization does not require as much representation change as learning the underlying structure present in the dataset, which is encoded as knowledge within the neural. From that, we conclude that if a model can obtain knowledge by learning the structure of the data, the earlier layers change more than if the network only memorizes.

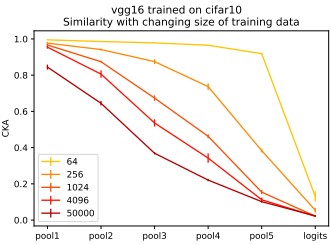 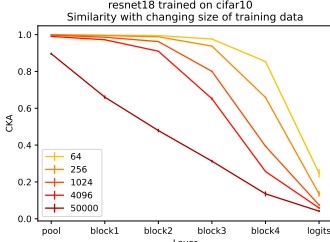

Figure 5: The figure shows the change of representation with respect to the numbers of samples in training size for random labels for Resnet18 and VGG16 trained on CIFAR-10.

In an additional experiment, we are interested in how the representations change with the size of the data when training on completely random labels. Especially in the case of the small-data regime (e.g., few-shot learning), only a few data samples are available. In Figure 5, we see that if there are only a few data samples, there is hardly any representation change in earlier layers. By increasing the training data size, the representation change increases in earlier layers used. So, even if the representation of earlier layers is a product of random transformations, the representation change is not as large as we have more data.

Overall, we can summarize that the representation change is larger in early layers if either the data has a better hierarchical structure (true labels) or if more data is available (data size).

## 5 DEEP NEURAL NETWORKS DURING FINE-TUNING

Instead of using randomly initialized weights, transfer learning leverages pre-trained weights. In this section, we analyze the representation change of models when they are fine-tuning. We use networks pre-trained on ImageNet and CIFAR-10 and fine-tune them on CIFAR-10, SVHN, and several randomized and shifted versions of it.

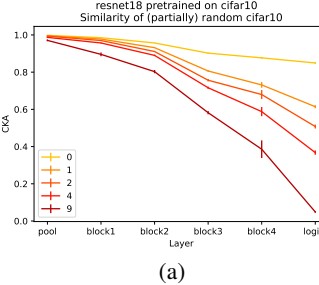 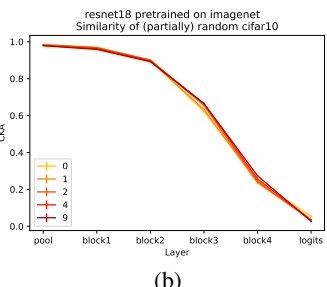

(a) (b)

Figure 6: The figures show the representation change of fine-tuned networks (ResNet18), which were initiated by pre-trained weights. The pre-training was done on both CIFAR-10 and ImageNet. Fine-tuning was applied to CIFAR-10 with several random degrees ($d \in \{0, 1, 2, 4, 9\}$).

### 5.1 (PARTIALLY) RANDOM LABELS

In this experiment, we want to model different situations. First of all, we compare fine-tuning the different random versions of CIFAR-10 on ResNet18, which all have been pre-trained on CIFAR-

10 (Figure 6(a)). We observe in Figure 6 the opposite of the results in standard deep learning (see Section 4.1). The more random the target data is, the more representation changes can be observed in all layers. The pre-trained weights on the very similar dataset make it unnecessary for the network fine-tuned on the structured data to change much. However, the knowledge of the network learned during pre-training is *unlearned* when there is no structure in the fine-tuned dataset. This is different from the results depicted in Figure 4. Applying the same fine-tuning on ResNet18 pre-trained on ImageNet (Figure 6(b)), we observe that the representation change is very similar for all layers. It shows that the learned representations of ResNet18 pre-trained on ImageNet change the same regardless of how the structure is in the fine-tuned data. The results of VGG16 are analogous to the ResNet's and are shown in Appendix C.

## 5.2 DOMAIN VS. CROSS-DOMAIN

In Figure 7, we want to compare the results of fine-tuning on domain (shifted) and cross-domain (SVHN) tasks. We know already from the previous section that representation change in early layers is larger than when fine-tuned on CIFAR-10 but significantly less than when fine-tuned on random labels, indicating that the models make use of pre-trained weights. Looking at domain adaptation (the shifted version of CIFAR-10), we see that for a problem within the same domain, the representation change is larger than the same-domain task but smaller than the random task as well as the cross-domain task (application on SVHN). We again see that for ImageNet, it is all the same because all the CIFAR-10 tasks are similar to the ImageNet problem.

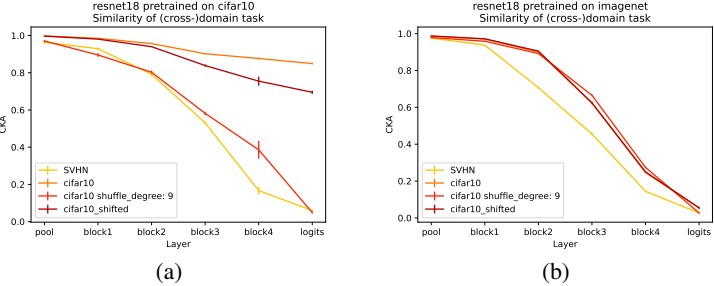

Figure 7: The figures shows the representation change of fine-tuned networks(ResNet18), which were initilized by pre-trained weights. The pre-training was done on both CIFAR-10 and ImageNet. Fine-tuning was applied to CIFAR-10, random CIFAR-10 ($d = 9$), CIFAR-10 shifted and SVHN.

## 5.3 PRE-TRAINED VS PRE-INITIALIZED

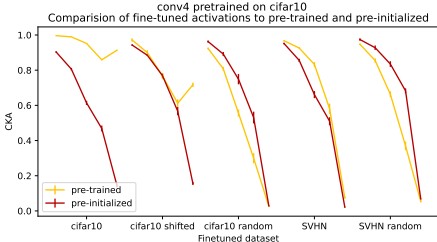

Figure 8: In this figure, we depict the comparison of the fine-tuned representation with the pre-initialized and the pre-trained representation of 4-conv pre-trained on CIFAR-10.

Our models are trained in two phases. In the first phase, it transforms the random weights into pre-trained ones, and in the second, it turns the pre-trained weights into fine-tuned ones. In these experiments, we compare the representation change of the fine-tuned weights to both the random initialization (*pre-initilized*) and the representation obtained by the *pre-trained* weights.

In Figure 8, we can observe that when pre-training a 4-conv network on CIFAR-10, the fine-tuned representations are closer to the pre-trained than to the pre-initialized one. This makes sense, as the task during pre-training was on data coming from the same distribution. Interestingly, when having completely random labels, this is different. The representations obtained after applying the final model to the data are not more similar to the pre-trained weights on which the fine-tune task started than the complete random initialization. This confirms our position that if the data is not meaningful and structured directly, it *unlearns* the learned knowledge, as they are not helpful in memorization. For the shifted dataset, we see that it uses the learned knowledge and is more similar to the pre-trained activations than to the pre-initialized ones. On SVHN, the difference is less but still significant, especially in the earlier layers, although the data of SVHN is not as similar to the pre-training dataset as CIFAR-10 random, which comes from the same distribution but has noisy labels.

Figure 9 shows the difference between pre-trained and pre-initialized activations for ResNet18 and VGG16 pre-trained on both ImageNet and CIFAR-10 and fine-tuned on CIFAR-10 and its random version[4]. In all fine-tuning tasks, we see that other than by training the 4-conv network, the representation change to the pre-training representation is significantly smaller than to the pre-initialized representation. It shows that larger models do not change as much as smaller models during fine-tuning, even if there is no meaningful hierarchical structure in the dataset (random dataset).

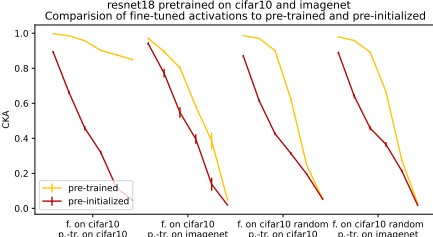 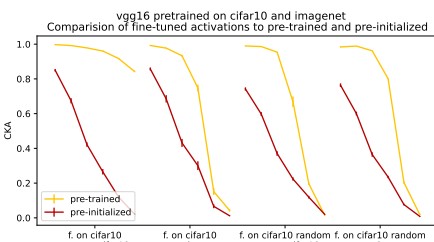

Figure 9: In this figure, we depict the comparison of the fine-tuned representation with the pre-initialized and the pre-trained representation of ResNet18 and VGG16 pre-trained on both ImageNet and CIFAR-10 and fine-tuned on CIFAR-10 and its random version ($d = 9$).

## 6 CONCLUSION

In conclusion, our paper contributes a nuanced perspective on the interpretation of representation changes in deep neural networks. We argue that the interpretation of small representation changes, especially in early layers, can be influenced by factors such as the dataset's hierarchical structure and the training sample's size. Therefore, we compare the representation change to experiments with randomized versions of our datasets. Our findings highlight that memorization causes less representation change in early layers, while they change more if encoding the knowledge of the structure of the data. Representation change in fine-tuning is larger in earlier layers if the fine-tuning task diverges from the pre-training task. Comparing representations to both its pre-trained weights and the pre-initialized ones, we find out that especially smaller networks tend to *unlearn* the pre-trained network if the fine-tuned dataset does not have a meaningful hierarchical structure. Overall, we want to assert the importance of careful interpretation. It is important to consider contextual comparisons with similar problems sharing the same architecture when assessing the significance of representation changes in neural network analysis.

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

## A EXAMPLE OF DATA MANIPULATION

Figure 10 depicts two examples of manipulating the CIFAR-10 dataset. One with a random degree of $d = 2$ and another with a shifted dataset.

## B FURTHER PERFORMANCE STATISTICS

In Figure 11, we deliver the training and test statistics of VGG16 and 4-conv applied on CIFAR-10.

## C FINE-TUNING EXPERIMENTS OF VGG16 AND 4-CONV

In Figure 12, we depict the results of fine-tuning VGG16 on the (partially random) datasets. In Figure 13, we can see the results of fine-tuning 4-conv on the (partially random) datasets.

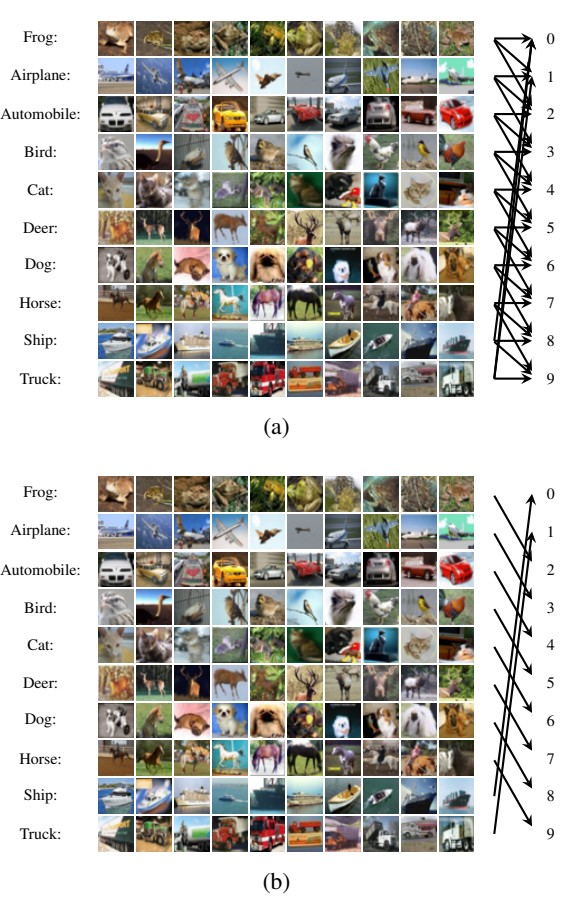

Figure 10: Randomized label assignment with a degree of $d = 2$ and shifted version of CIFAR-10.

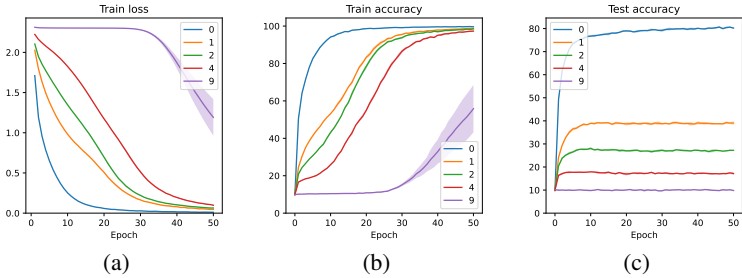

(a)            (b)            (c)

Figure 11: This figure depicts the training loss, training accuracy, and the accuracy during testing of the application of VGG16 for degrees of randomness $d \in \{0, 1, 2, 4, 9\}$ of CIFAR-10.

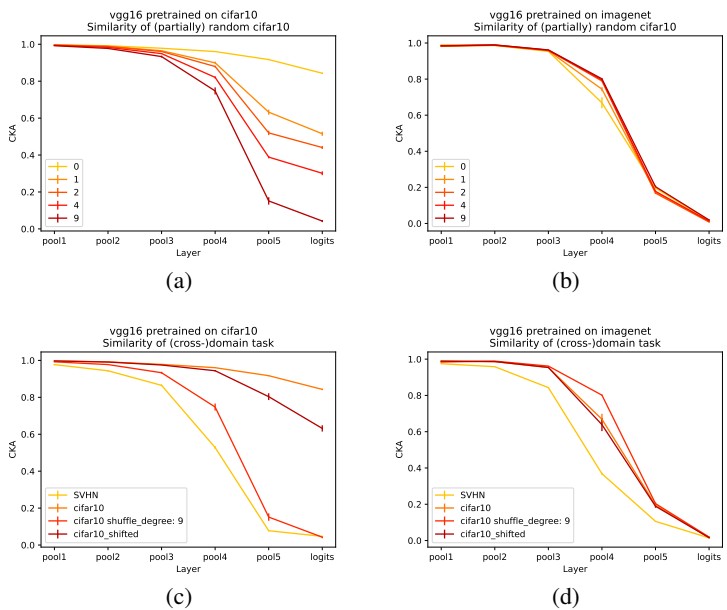

Figure 12: The figures show the representation change of fine-tuned networks (VGG16), which were initiated by pre-trained weights. The pre-training was done on both CIFAR-10 and ImageNet. Fine-tunding was applied to CIFAR-10 with several random degrees ($d \in \{0, 1, 2, 4, 9\}$ and to CIFAR-10, random CIFAR-10 ($d = 9$), CIFAR-10 shifted and SVHN.

# D   FURTHER COMPARISION OF REPRESENTATIONS OF PRE-INITIALIZATION AND PRE-TRAINING

In Figure 14, we show further fine-tuned results of VGG16 and ResNet18, where we compare pre-initialization with pre-training.

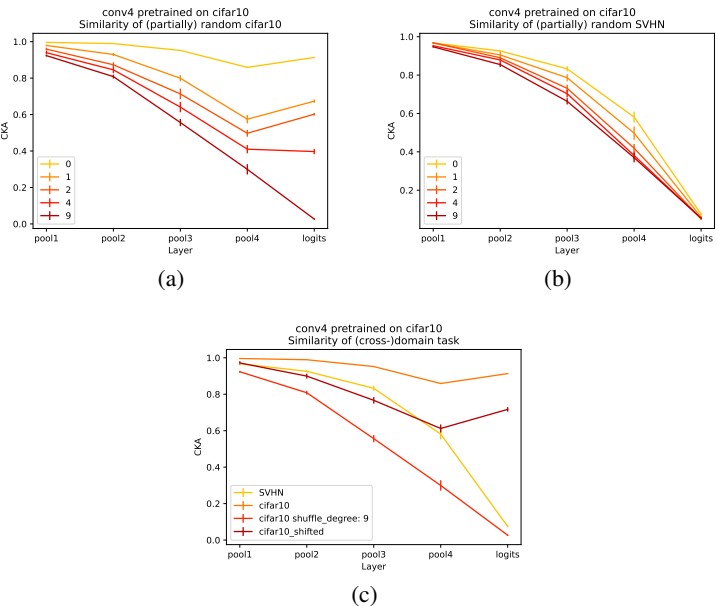

(a)      (b)

(c)

Figure 13: The figures show the representation change of fine-tuned networks (4-conv) initialized by pre-trained weights. The pre-training was done on CIFAR-10. Fine-tunding was applied to CIFAR-10 with several random degrees ($d \in \{0, 1, 2, 4, 9\}$, SVHN with several random degrees ($d \in \{0, 1, 2, 4, 9\}$ and to CIFAR-10, random CIFAR-10 ($d = 9$), CIFAR-10 shifted and SVHN.

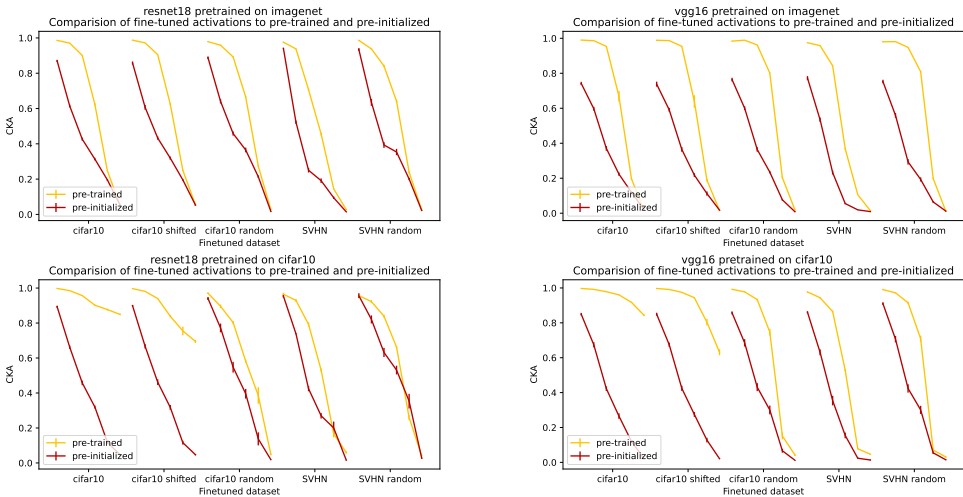

Figure 14: In this figure, we depict the comparison of the fine-tuned representation with the pre-initialized and the pre-trained representation of VGG16 and ResNet18 pre-trained on both ImageNet and CIFAR-10.

