# OpenReview forum: "Unveiling the Dynamics of Transfer Learning Representations"
_ICLR.cc/2024/Workshop/Re-Align — ICLR 2024 Workshop Re-Align Poster_

### Official Review · Reviewer_WnXc · 2024-02-23
**Submission22 fails to clearly layout conclusions and novelty**

**Rating:** 1
**Fit:** 3
**Confidence:** 3

**Workshop Review:**

Submission22 aims to understand representational changes in different layers during transfer learning. They use different levels of randomization of training labels to induce loss of structure in their networks as control, and find that earlier layers have less representational change than later layers.

First, the writing in the paper makes it difficult to assess the main arguments and novelty of the findings that the authors want to stress. Sentences do not seem to connect well to each other.
In general, it is unclear what novel contributions the paper is making. The fact that early layers learn more general features than later layers is well known, as stated by the authors themselves. While there may be some discussion and nuance in how useful transfer learning of the last layer is, the authors do not discuss or cite these papers. The use of randomization, while the authors claim they have a novel "gradual" randomization, is fairly simple and not that exciting. Lastly, the idea that networks that were trained on complete nonsense, when then presented with actual structure data, "unlearn" their initial representations, is both not surprisingly and slightly misleading. How can representations be unlearned when they were in fact, not learned in the first place?
The use of only one metric for representational similarity (which is also not novel and never formally defined in the main text) in all figures is limiting.

References:
[1] Ananya Kumar, Aditi Raghunathan, Robbie Matthew Jones, Tengyu Ma, and Percy Liang. Fine-tuning can distort
pretrained features and underperform out-of-distribution. In International Conference on Learning Representations,
2022. URL https://openreview.net/forum?id=UYneFzXSJWh.
[2] Elad Hoffer, Itay Hubara, and Daniel Soudry. Fix your classifier: The marginal value of training the last weight layer.
arXiv preprint arXiv:1801.04540, 2018.
[3] Frati, Lapo, et al. "Reset It and Forget It: Relearning Last-Layer Weights Improves Continual and Transfer Learning." arXiv preprint arXiv:2310.07996 (2023).

**Reason For Not Giving Higher Score:**

The paper is poorly written and lacks novelty.

**Reason For Not Giving Lower Score:**

N/A

**Reviewer Domain:**

neuroscience

---

### Official Review · Reviewer_WKkB · 2024-02-23
**Clearly work in progress, but worth accepting :)**

**Rating:** 2
**Fit:** 3
**Confidence:** 2

**Workshop Review:**

Accept:

This paper passes the standards for a workshop paper and is clearly relevant to the workshop.

Issues:

Not sure how much I believe the experiments generalize or are informative beyond CIFAR.

Minor Edits:

Figures are very hard to read because of the similar colors. Please use colors that are less close together AND use unique tics for each line.


PAPER SUMMARY:

Authors conduct small scale experiments probing representation change over training and fine-tuning using a) controlled randomization of the dataset and b) modifying the amount of the data.

The high level message is that interpreting the amount of layer change is very context dependent on the details of the data and architecture.

This is a clear and unsurprising message.   A key weakness of the paper is I don’t find it surprising.

To be specific, in this paragraph `We analyze representation obtained by models having initialized, pre-trained, and fine-tuned weights. By analyzing the training pre-training deep network model, we confirm our position that early layers structurally change less than later ones and that a small fine-tuned dataset reduces the change of early layers additionally (Section 4). To investigate the representation change during fine-tuning, we apply pre-trained models to structured, unstructured, domain, and cross-domain tasks. We confirm that the data structure is encoded in the early convolutional layers. In addition, we reveal that if the transfer task is unstructured, the pre-trained information is unlearned in smaller networks but used in larger networks, whereas, in a domain or cross-domain adaption, it exploits it also for smaller networks (Section 5).’

This paragraph summarizes the contributions of the paper. I would argue that this is all pretty typical with standard understanding of deep learning, as observed in the previous work by Raghu and Oh.

To make this paper stronger, I think the authors need to further explicate why this false intuition about layer wise change is harmful. Specifically point out methods or interpretations that are incorrect and falsified by these experiments?

While interesting, this work does not suggest algorithmic improvements, nor do I see how it can lead to algorithmic improvement. That said, it is worth falsifying a common assumption, or trying to do so.

I would also like the authors to expand on how to rectify these issues on representation change. What properties would a better method have?

Clarity: Moderate, lots of imprecise and vague prose.

Correctness:  Strong-I looked at the code briefly and it looked mostly fine. I am skeptical these results will meaningfully scale to larger architectures/harder tasks.

Novelty:  Questionable. I feel like very similar randomization experiments have been conducted or are folklore.

**Reason For Not Giving Higher Score:**

See above but the big reasons are a) small scale of experiments b) unclear, high level message c) unclear foundation for future research. What is the bottom line of the research? What should I do differently when fine-tuning? What could be a better method for interpreting changes in layers? Should that line of intepretability be abandoned?

The authors do not take strong positions on any of this topics and are somewhat vague when answering these questions.

**Reason For Not Giving Lower Score:**

The authors do critique an existing baseline/practice-so there is some clear novelty/tension with current practice.

**Reviewer Domain:**

machine learning

---

### Official Review · Reviewer_CwgM · 2024-02-23
**Solid work relevant to representational alignment**

**Rating:** 3
**Fit:** 3
**Confidence:** 2

**Workshop Review:**

This paper argues for the importance of comparing a network architecture's hidden representations across datasets, rather than merely comparing representations across layers within an architecture. The authors propose a method for varying how much structure is present in a dataset by shifting labels to varying degrees, such that with more degrees there is less structure in the labels. This forces the network to rely on memorizing data points rather than generalization, since with the highest degree of randomness, labels become meaningless. They demonstrate a minimal case where the representations of a DNN's deepest layers shift more greatly than the shallower layers on a shifted-labels dataset, and conclude that memorization is happening more in later layers than early layers. This trend is modulated by label randomness and dataset size, with less random labels and more training data leading to greater changes in early layers. The authors modify this domain to more closely resemble transfer learning by comparing networks which are initialized, pre-trained, and fine-tuned on different datasets, and find that models "unlearn" weights when labels are randomized, similar to transferring across two fairly different domains, CIFAR and SVHN.

I find the author's results and arguments convincing, and highly relevant to this workshop. The experiments are thorough and results are presented well, and the dataset transformation methods are well-explained. I think the examples of Raghu & Oh provided in the introduction are effective in grounding the significance of this work. I find the result that later layers change more than early layers across datasets, and for memorization over generalization, interesting and fairly intuitive.

I think that by and large this is well presented and well-written, although the writing in a few sections left me a little confused about some key results.
- 5.1: "We observe that the opposite of the results in standard deep learning happens (see Section 4.1). The more random the data is, the more the representation changes. This is caused by the fact that we have pre-trained weights that are trained to solve the task and not random weights."   I don't understand this point. Perhaps I'm misunderstanding something, but it seems to me that a standard deep learning view of this setup would predict more change in representation with more randomness because of greater distribution shift between pre-training and fine tuning.
- I think the patterns in Figure 6 across left/right are striking, though I wish I understood exactly what the authors' interpretation of these are. It's surprising to me that the representation similarity trend across imagenet/cifar is most similar to cifar/randomized-cifar, I'd expect that since there is a lot of similar information in imagenet+cifar, to generalize instead of memorize, so the pattern would be the opposite of what you observe (imagenet+cifar more similar to d=0).
- Are Figures 6 and 7 referred to in the text anywhere?
- Some of the figure labels seem off, e.g. "Figure 5.2" in Section 5.2, or "Figure D".

Smaller point
- There's a couple typos: "tunding", "unleared"

**Reason For Not Giving Higher Score:**

N/A

**Reason For Not Giving Lower Score:**

Key arguments, methods, and results are novel, thorough, and (for the most part) well-presented. Arguments are convincing based on their results. The significance of this work is well grounded in related prior work.

**Reviewer Domain:**

cognitive science

---

### Decision · Program_Chairs · 2024-03-02

Accept (Poster)